# Immunoprotecting Effects of Exercise Program against Ovarian Cancer: A Single-Blind, Randomized Controlled Trial

**DOI:** 10.3390/cancers14112808

**Published:** 2022-06-05

**Authors:** Jong-Kyun Lee, Sihwa Park, Yong-Seok Jee

**Affiliations:** 1Research Institute of Cell Immunity for Cancers, Seoul Songdo Hospital, Seoul 10083, Korea; leejksd@gmail.com; 2Research Institute of Sports and Industry Science, Hanseo University, Seosan 31962, Korea

**Keywords:** regular exercise, ovarian cancer, muscle mass, NK cells, NK cell receptors

## Abstract

**Simple Summary:**

The impact of exercise on the immune system is pleiotropic, and our knowledge about how exercise-triggered mechanisms affect cancer risk and progression is incomplete. Although several mechanisms have been proposed to explain the protective effect of exercise against cancer, there is a lack of experimental evidence in cancer survivors, which is related to immunity. In addition, no specific mechanism has been proposed for exercise-induced suppression of cancer cells through improving innate immunity in ovarian cancer survivors. Therefore, the aim of this study was to investigate the effects of regular exercise training on physical fitness, including body composition and innate immunocytes, in ovarian cancer survivors.

**Abstract:**

Exercise is known to help the immune function of cancer survivors after cancer cell removal, but there is little information about the effect of exercise on ovarian cancer survivors. We conducted this study to investigate the effects of exercise training on the physical fitness and innate immunity of ovarian cancer survivors (OCS). Twenty-seven OCS between forty-two and sixty-one years of age volunteered for this study. The participants were divided into a control group (COG, *n* = 15) and an exercise group (EXG, *n* = 12). The mean (SD) age was 51.07 (5.67) years, and the mean post-operation period was 45.96 (5.88) months. EXG participated in regular exercise training 6 days a week for 12 weeks. Body weight, fat mass, and body mass index of EXE were significantly decreased compared with those of COG. The muscle mass in EXE was increased compared to that of COG. Physical fitness factors showed positive changes in EXG compared to COG. We found that exercise training enhanced lymphocyte and neutrophil counts of leucocytes and total natural killer (NK) and natural killer T (NKT) cell counts of lymphocytes through improved body composition and physical fitness after 12 weeks. Moreover, we found that improved innate immune cells through the exercise program were achieved through an increase in NKG_2_D+NK receptors and a decrease in KIR_2_DL_3_+NK receptors in OCS.

## 1. Introduction

Cancers are diseases dominated by uncontrolled cell growth that results in the compression, invasion, and degradation of surrounding tissue. Malignant cells can be transported through the blood or lymphatic fluid to peripheral tissues or organs and cause secondary colonies [1]. Cancer is one of the primary causes of premature death. Ovarian cancer occurs due to abnormal cell growth in the ovaries and is the most lethal gynecological tumor, accounting for 4.4% of cancer-related deaths worldwide [2]. The five-year survival rate of epithelial ovarian cancer patients varies depending on the stage of the tumor [3]. It declines from 89% at stage I to 20% at stage IV, but diagnosis occurs at the advanced stages (III or IV) in 80% of patients, as specific symptoms of the disease are not apparent [4].

Chemotherapy and cytoreductive surgery are the standard forms of treatment for ovarian cancer [5]. The symptoms of cancer are diverse and depend on tumor type and locality. When ovarian cancer first develops, it might not cause any noticeable symptoms. When ovarian cancer symptoms occur, they are usually attributed to abdominal bloating, satiety, constipation or frequent urination, pelvic discomfort, low back pain, fatigue, and weight loss [6]. Many types of cancer, including ovarian cancer, cause sarcopenia as well as fatigue and reduced physical fitness [7]. The discomfort and poor appetite that cancer survivors experience from undergoing treatments, such as chemotherapy, radiotherapy, and surgery, lead to physical inactivity, which has been estimated to account for 25–50% of cancer patients’ poor physical condition and sarcopenia [1,8]. This deconditioning in cancer survivors leads to decreased immunocytes that destroy malignant cells. However, research has shown that regular exercise can improve the physical fitness of cancer patients and survivors, thereby fighting against deconditioning and enhancing immunity [9].

A critical corollary is whether exercise impacts the antitumor efficacy of cancer treatment. This notion is biologically plausible, as emerging evidence suggests that exercise modulates several factors inherent in cancer treatment sensitivity. Arguably, some of the most relevant are alterations in the tumor microenvironment [10]. There are many positive results from exercise training, but the most critical one is producing immunocytes that can attack cancer cells that may emerge in cancer patients. It is important to preserve and nurture immunity so that cancer does not recur or metastasize in those who survived cancer removal. Among many immunity-related cells, natural killer (NK) cells have cytotoxic activity against malignant cells and play a key role in antiviral and tumor immunity [11,12,13]. These recognize stress signals, cancer transformation, or infection with immediate effectors achieved via expressing of a wide array of receptors, such as NK cell-activating receptors, NKG_2_D (natural killer group 2, member D), and NK cell-inhibiting receptors, KIR (killer cell immunoglobulin-like receptor). In addition to cytotoxic elimination of target cells, they are potent producers of cytokines and chemokines and can promote an innate immune response. The pleiotropic functions and the remarkable effector agility of NK cells make them ideal candidates for immunotherapeutic interventions [14].

Exercise activates the sympathetic nervous system and causes a disturbance in the endocrine system [15,16]. Epinephrine is released into the blood, promoting NK cells that destroy compromised host cells, such as tumors or virus-infected cells. Regarding the above study results, a 45–60-minute bout of vigorous exercise increases NK cell concentrations 10-fold [16]. Previous studies have also reported that specific exercise training enhances the immune system [9,17]. In other words, growing epidemiological evidence has represented that a physically active lifestyle shelters against the induction of various cancers [18,19]. Several observational studies have shown that people who are physically active after being diagnosed with breast cancer, colon cancer, and others have a significantly higher survival rate compared to those who are not physically active [20,21,22,23]. Although there are many studies on the benefits of exercise in patients or survivors of breast, colon, prostate, thyroid, and other cancers, there are few studies for ovarian cancer patients or survivors. In particular, the effect of exercise regimens may be more important in extending the lifespan of the survivors rather than cancer patients; however, research related to this is also lacking.

The impact of exercise on the immune system is pleiotropic, and our knowledge about how exercise-triggered mechanisms affect cancer risk and progression is incomplete [24]. Although several mechanisms have been proposed to explain the protective effect of exercise against cancer, there is a lack of experimental evidence in cancer survivors, which is related to immunity. In addition, no specific mechanism has been proposed for exercise-induced suppression of cancer cells through improving innate immunity in ovarian cancer survivors. Therefore, the aim of this study was to investigate the effects of regular exercise training on physical fitness, including body composition and innate immunocytes, in ovarian cancer survivors.

## 2. Materials and Methods

### 2.1. Participants

The participants of this study were ovarian cancer survivors who were diagnosed with ovarian cancer between January 2016 and December 2020 and who underwent interval debulking surgery, primary cytoreductive surgery, radiation therapy, or chemotherapy, with a survival rate of less than five years. The inclusion criteria, recruitment, and study design for the study have been previously reported [9]. Considering the purpose of this study, survivors were subjected to continuous nutritional management at the hospital, but only subjects who had no experience participating in a regular exercise program. All participants who volunteered for this study consisted of ovarian cancer survivors between 42 and 61 years of age. Subjects who had severe arthritis, obesity, psychiatric illness, uncontrolled hypertension, or problems with a major organ system that could interfere with exercise were excluded from the study.

Initially, 37 participants were screened to determine their study eligibility. In the process of classifying patients, three participants had to be excluded due to severe obesity and arthritis. Therefore, 34 participants were assigned using random number tables and given identification numbers upon recruitment. They continued to receive standard care, with the agreement that the existing physical activity volume was to remain unchanged over the duration of the trial. Participants were informed of their allocation by a researcher and were randomly allocated to either the control group (COG, *n* = 15) or the exercise group (EXG, *n* = 12). At any stage, all assessments were performed at sites from the hospital, and participants were reminded before each assessment not to disclose their allocation. Two participants in the COG dropped out during the allocation phase, and five subjects in the EXG declined during the allocation, follow-up, and analysis phases. Finally, a total of 27 subjects were enrolled into the study, as shown in Figure 1.

All the participants were admitted to the hospital from Monday to Saturday, received intensive care for exercise, and went home only on Sunday. Survivors in both groups ate meals prescribed by a nutritionist at the hospital from Monday morning to Saturday afternoon during the three months of the experiment. As for the type of diet, macronutrients such as carbohydrates, proteins, and fats and micronutrients such as vitamin A and vitamin C were ingested, and one of the researchers analyzed the recorded diet once a week. To prevent communication between the groups, the participants were classified according to their inpatient ward. The interventions and measurements were also scheduled at different times: COG at 10:00 and EXG at 14:00. As opposed to EXG patients participating in exercise, COG patients gathered at the same center at a scheduled time and were allowed to meditate in a lying position on a mattress for 60 min. At this time, the COG patients were allowed to lightly stretch in a lying position on the floor but did not engage in strength training or other aerobic exercise, while sleeping when they were drowsy.

### 2.2. Experimental Design

This single-blind, randomized controlled trial was conducted from 4 October to 24 December 2021. Ovarian cancer survivors and their family members were given motivation and encouragement to complete the tests and the regular exercise regimen. Advertisements were used to recruit volunteers who signed an informed consent form prior to enrolling and starting the experiment. This study followed the principles of the Declaration of Helsinki (2013 version) and received approval from the institutional ethics committee of Seoul Songdo Hospital (27 August 2021 to 26 August 2022; 2021-008). This study was registered at the Korean Clinical Research Information Service (KCT PRE20220508-001). During the pre-experiment session, a diary was provided to each subject for recording all their food consumption during the study. An expert input the food type and volume in CAN-Pro 5.0 (Korean Nutrition Society, Seoul, Korea) every day, calculated the caloric intake, and then performed an evaluation at the end of each week.

The daily amount of physical activity was also recorded and calculated using a shortened version of the international physical activity questionnaire (IPAQ) [25]. Each day, an expert distributed a diary that contained questions for the subjects to answer based on their weekly physical activities for the duration of the experiment. The daily calorie output was calculated by metabolic equivalent minutes (kcal/kg/min). The total calories were obtained through the summation of the duration (in minutes) and frequency (days) of walking (3.3 × min of activity/day × days per week), moderate-intensity activity (4.0 × min of activity/day × days per week), and vigorous-intensity activity (8.0 × min of activity/day × days per week). Then, using the data, the average amount of physical activity per week was calculated based on the IPAQ score conversion method [26]. These data were averaged and analyzed on a weekly basis.

### 2.3. Measurement Methods

#### 2.3.1. Blood Sample Collection and Measurements of NK Cell-Related Factors

We obtained peripheral blood using heparinized collection tubes and EDTA tubes. Whole blood samples were used for the automated differential blood cell counts and fluorescence-activated cell sorting (FACS) analysis. EDTA-anticoagulant blood samples for automated leucocyte differential tests were submitted to our hematology laboratory and tested on the Sysmex XN-550 (Sysmex Corporation, Kobe, Japan) as the primary routine method. The samples were stored at room temperature no longer than 4 h prior to testing. A lysed whole blood technique with maximal eight-color staining of blood samples was used for flow cytometry and antibody staining, as shown in Figure 2.

Subpopulations of human peripheral blood cells were analyzed as follows. Blood 50 μL was stained with anti-human antibodies against anti-human CD3-Fluorescein isothiocyanate (FITC; Cat No. 555339), CD56-Phycoerythrin (PE; Cat No. 555516), Human NKG2D/CD314 Antibody (Cat No. MAB139-SP), and KIR2DL3/CD158b (Cat No. MAB2014-SP) from BD Biosciences, together with appropriate isotype controls. All antibodies were obtained from BD Biosciences (Franklin Lakes, NJ, USA). Then, 30 min of incubation took place at room temperature in a dark environment before adding 500 μL of FACS lysing solution (BD Biosciences) to erythrocytes in each test tube for 15 min. Then, 2 mL of permeabilization buffer (DPBS 1X (WELGENE) was used to wash the remaining cells. After completing the process of staining, FACS Canto II (BD Bioscience) and Flowjo software (Treestar, Ashland, OR, USA) were used to analyze the cells. The data are presented as percentages. Finally, NK cell function was measured using an enzymatic colorimetric assay involving LDH release (CytoTox^®^ Non-Radioactive Cytotoxicity Assay kit, Promega) per the manufacturer’s protocol. The percentage of cytotoxicity was calculated with the following formula:Percent cytotoxicity = 100 × (Experimental LDH release)/(Maximum LDH release)

#### 2.3.2. Measurement of Physical Fitness

The physical fitness factors were body composition, cardiorespiratory endurance, muscle endurance, muscle strength, and flexibility. Prior to the body composition test, we measured height using an extensometer. Then, body composition was measured using a segmental impedance device that assessed the voltage drop in the upper and lower body (Inbody 770, Bio-space Co., Ltd., Seoul, Korea). Before stepping onto the platform, the subjects were instructed to remove any material that can cause electrical interference. They were also given instructions to stand still while holding onto the handles for 3 min. To minimize error, the participants were asked to avoid food intake for 4 h, alcohol consumption for 48 h, and any kind of exercise for 10 h prior to the test. They were then asked to void 30 min before the test [27].

A graded exercise test assessed peak oxygen uptake (VO_2_ peak) for cardiorespiratory endurance. A treadmill ergometer (Q65-90, Quinton, OK, USA), electrocardiogram (Q-4500, SunTech Medical, Inc. Morrisville, NC, USA), automatic sphygmomanometer (M-412), and gas tester (QMC4200) were used. The electrodes were attached to the chest cavity, and the blood pressure cuff was placed on the left arm. The mouthpiece was fixed over the lip and nose area. The modified Bruce protocol was used for the ovarian cancer survivors who continued walking to running until the maximum limit was reached, which was their peak rating of perceived exertion (RPE). During and after walking or running for as long as possible, the participants were asked to describe their symptoms, such as chest pain, leg pain, breathlessness, and/or dizziness. If the participants wished to discontinue the test, the test was terminated. The measure was also ceased if: (a) ≥10 mmHg drop in systolic blood pressure from baseline, (b) moderate-to-severe angina, (c) an increase in neural symptoms, (d) cyanosis, (e) sustained ventricular tachycardia, (f) >1 mm ST elevation in leads without diagnostic Q waves, and (g) technical difficulties in monitoring electrocardiographic tracings were represented [28].

Grip strength for muscle strength was measured using a Smedley dynamometer. The hand and the forearm muscles were included in the test. The participants held the dynamometer so that the proximal interphalangeal joint formed a right angle, and the width was adjusted accordingly. The participants’ arms were placed at their sides in a natural position with the dynamometer not touching the body. Both hands were consecutively assessed twice, and the maximum value was recorded [26]. The strength values of both arms were averaged and analyzed.

Sit-ups for muscular endurance were measured. The participants laid down with their backs against a mat, their knees were bent at 90°, and their feet were 30 cm apart while putting their ankles on the footrests. Both hands were locked in front of the chest. When the examiner gave the signal to start, the participants curled their upper body forward so that their elbows touched their knees. Both elbows had to reach the knees, and the back had to reach the floor for the movement to be counted as one repetition. This sit-ups test, completed in 60 s, was recorded [29].

A trunk forward flexion test was used to assess flexibility. Before the assessment, subjects completed a standardized warm-up that included 5 min of stretching exercises. A flexibility measuring device, TKK1859 (Takei Inc., Tokyo, Japan), was used to measure trunk flexion through the sit-and-reach test. Participants were told to fully extend their legs while keeping their knees relaxed before extending their hands forward as far as possible and maintaining the position for 2 s [30]. The greatest value after two measurements was recorded.

### 2.4. Regular Exercise Training Program

The EXG participated in a supervised exercise program six times a week for twelve weeks. The dosage of exercise was determined through the FITT principle, wherein the frequency, intensity, time, and exercise type quantify the total dose. We referred to specific recommendations for physical activity dosage for cancer prevention. When designing an exercise program, we prescribed 150 min of moderate-intensity or 75 min of vigorous-intensity exercise per week, as suggested by the American Cancer Society and the American College of Sports Medicine [31,32].

As shown in Table 1, the EXG began by warming-up for 10 min, as instructed by a researcher, and then performed the scheduled workout. In the case of aerobic exercise training, participants performed aerobic exercise three days a week (Mondays, Wednesdays, and Fridays), and the intensity of exercise was walking or running at 40–70% of the peak oxygen uptake. A heart monitor was used to maintain heart rate according to the exercise intensity. The exercise started with 50 min of walking in Phase I and 35 min of brisk walking to light jogging in Phase IV, after which the program ended.

In this study, a resistance exercise capable of lifting 12 repetitions maximum (RM) to 6 RM was applied to cancer survivors based on the previous studies [9,33,34]. In detail, the resistance exercise was performed at 12 RM for Phase I, 10 RM for Phase II, 8 RM for Phase III, and 6 RM for Phase IV, respectively. The EXG participated in a resistance exercise for three days per week (Tuesdays, Thursdays, and Saturdays). For the safety of the ovarian cancer survivors, exercise using weight machines was prescribed, and according to exercise order, multiple-joint exercises centered on large muscles were performed. Afterward, single-joint exercises were carried out in parallel. The order of resistance exercise consisted of lateral pull-downs, chest presses, leg presses, abdominal crunches, and back extensions on Tuesdays, seated cable rows, chest butterflies, deadlifts, leg extensions, and leg curls on Thursdays, and lateral pulldowns, chest presses, leg presses, abdominal crunches, and back extensions on Saturdays. The reason why the exercise was prescribed differently for every other day was to ensure that the patients had no experience in exercising regularly and that the exercise they encountered for the first time did not put any strain on the muscles and tendons. All exercises were performed under strict supervision, and a RPE was applied to each exercise to identify the fatigue from the exercise and then adjust the rest time. The EXG performed the resistance workout sessions for 30–40 min. After these exercises, 15 min stretching was performed in sitting and lying positions. Total workout time per day (for 3 days) was 80–90 min (240–270 min) in Phase I, 75–85 min (225–255 min) in Phase II, 70–80 min (210–240 min) in Phase III, and 65–75 min (195–225 min) in Phase IV.

Some exercises were contraindicated by recent blood test results or physical conditions, and these limitations were strictly adhered to. Survivors undergoing chemotherapy or radiotherapy with a leukocyte count below 0.5 × 10^9^/L, hemoglobin count below 6 mmol/L, thrombocyte count below 50 × 10^9^/L, or temperature above 38 °C did not perform such exercises [35]. Survivors with bone metastases avoided resistance training with heavy weights. In case of infection, it is recommended that training be interrupted for at least one whole day without symptoms, after which time training should be slowly resumed. In addition to the guidelines presented above, this study not only encouraged survivors of ovarian cancer to perform well, but also guided them by monitoring their physical condition during exercise.

### 2.5. Sample Size Measure and Statistical Analysis

Using a mean difference (and standard deviation, SD) between the groups with relevant research results on exercise interventions, the samples for a general linear model were calculated with α = 0.05, β = 0.95, and effect size = 0.40 through G*Power 3.1.9.7 [36,37]. At least 18 participants were required, which was satisfied by the number of subjects who took part in this study. The calculated samples were divided into an intervention group and a control group with the same number of people. The software estimated that nine survivors were needed per group. The statistical analysis was carried out with the GraphPad Prism 9.3.1 software (Graph Pad Software, San Diego, CA, USA), with a level of significance set at α = 0.05. The normality of all measured variables was analyzed using the D’Agostino and Pearson omnibus normality test. When comparing the mean values in normally distributed populations of quantitative data, Student’s t test was calculated. The Mann–Whitney U test was used to compare two independent groups when there was no evidence of normal distribution. The Wilcoxon signed rank test was used to analyze the statistical significance of differences in quantitative characteristics for the two dependent samples. The delta percentage (Δ%) analysis was conducted to compare the changes between times, and significance was also tested on this value with the Mann–Whitney U test.

## 3. Results

### 3.1. Demographic and Clinical Characteristics

Twenty-seven participants were enrolled from 1 to 30 September 2021. Of the 37 survivors approached, 27 participated in the study. Baseline survivor characteristics are shown in Table 2. Median age was 53 (range 42–61). Based on the surgical stage of the International Federation of Gynecology and Obstetrics, ovarian cancer removal was performed in 3 patients at stage I, 13 at stage II, 6 at stage IIIA, and 5 at stage IIIB. Most of the patients with ovarian cancer underwent primary cytoreductive surgery (*n* = 20, 74.1%), and the rest underwent interval debulking surgery (*n* = 7, 25.9%). Age, height, body mass index, and clinical characteristics of the ovarian cancer survivors between both groups were not significantly different prior to the experiment.

### 3.2. Similarity of Control Variables before and after the Experiment

Table 3 shows the controlled variables of this study, such as daily calorie intake and physical activity. The daily calorie intake of the two groups was almost 1600 kcal/day, and there was no significant difference before the experiment. Similar results appeared in the two groups after the experiment. In particular, the daily intake of carbohydrates and fat, which are the main energy sources during exercise, was similar between the two groups, and the intake of protein important for immune cell function was 50.20 ± 6.28 g/day for the COG and 51.17 ± 6.95 g/day for the EXG before the experiment. After the experiment, the protein intake was 51.67 ± 5.61 g/day for the COG and 50.67 ± 7.88 g/day for the EXG, indicating that both groups ate similarly throughout the experiment period. Meanwhile, the COG and EXG were requested to limit factors of daily activities other than the exercise program provided in this study for 12 weeks. The COG was 264.47 ± 27.83 kcal/day, whereas the EXG was 276.00 ± 10.37 kcal/day before the experiment. This daily physical activity level, calculated and averaged on a weekly basis, appeared similar to that before the experiment even at the end of the experiment, and there was no significant difference between groups.

### 3.3. Positive Changes of Physical Fitness Levels

A good physical fitness level maintains a good human body composition and brings about desirable changes in immune function. It can be inferred that this knowledge is more important to survivors who need to extend their lifespan than healthy people. However, there are still insufficient studies on whether exercise training provides a positive effect for cancer patients or survivors. For this purpose, this study obtained the following results of performing exercise 6 days a week for 12 weeks in ovarian cancer survivors. As shown in Figure 3A, VO_2_ peak decreased in the COG (33.48 ± 5.43 to 30.10 ± 4.68 mL/kg/min, −8.89% ± 14.43%; Z = −2.651, *p* = 0.008), but increased in the EXG (35.49 ± 2.62 to 39.22 ± 2.15 mL/kg/min, 11.02% ± 9.28%; Z = −2.364, *p* = 0.018), indicating a significant difference between groups (Z = −3.816, *p* = 0.001). Flexibility, as shown in Figure 3B, was significantly decreased in the COG (12.67 ± 2.54 to 11.09 ± 2.93 cm, −10.47% ± 23.60%; Z = −1.989, *p* = 0.047) but increased in the EXG (13.00 ± 2.30 to 18.08 ± 3.99 cm, 40.28% ± 27.86%; Z = −2.946, *p* = 0.003), indicating a significant difference between groups (Z = −3.614, *p* = 0.001).

After 12 weeks of exercise training, as shown in Figure 3C, muscle strength decreased significantly in the COG (31.19 ± 6.36 to 27.75 ± 5.68 kg, −10.12% ± 12.21%; Z = −2.820, *p* = 0.005), but increased significantly in the EXG (32.90 ± 3.37 to 41.82 ± 4.45 kg, 28.32% ± 18.68%; Z = −2.984, *p* = 0.003), indicating a significant difference between groups (Z = −4.206, *p* = 0.001). In the case of muscular endurance, as shown in Figure 3D, the results were similar to those of muscle strength. In other words, while the muscular endurance of the COG (6.40 ± 1.99 to 5.00 ± 2.85 reps., −22.13% ± 35.51%; Z = −2.063, *p* = 0.039) showed a tendency to decrease, the muscular endurance of the EXG (6.58 ± 1.98 to 13.25 ± 2.49 reps., 126.54% ± 101.23%; Z = −2.940, *p* = 0.003) was greatly improved and showed significantly different results after the experiment, indicating a significant difference between groups (Z = −4.191, *p* = 0.001), so it was found that the exercise program in this study resulted in a greater development of endurance.

### 3.4. Body Composition Changes by a Decreased Fat Mass and by an Increased Muscle Mass

To determine whether regular exercise training in the EXG plays a role in the change in body composition for 12 weeks, aerobic and strength training were performed every other day. Although body composition did not show a significant difference between the two groups before the experiment, after 12 weeks, body weight, as shown in Figure 4A, showed a tendency to increase in the COG (57.14 ± 6.13 to 59.67 ± 7.68 kg, 4.58% ± 9.18%; Z = −1.968, *p* = 0.049), whereas a tendency to decrease in the EXG (57.41 ± 13.33 to 53.92 ± 10.52 kg, −5.24% ± 5.98%; Z = −2.511, *p* = 0.012), indicating a significant difference (Z = −1.369, *p* = 0.183) between the groups after the experiment. Fat mass, as shown in Figure 4B, increased in the COG (18.90 ± 3.74 to 22.11 ± 5.08 kg, 19.68% ± 26.97%; Z = −2.275, *p* = 0.023) but decreased in the EXG (18.54 ± 5.12 to 16.57 ± 4.15 kg, −8.78% ± 16.08%; Z = −2.044, *p* = 0.041) after 12 weeks, indicating a significant difference between groups (Z = −2.932, *p* = 0.003). Body mass index (BMI), as shown in Figure 4C, increased in the COG (23.61 ± 3.03 to 24.95 ± 3.43 kg/m^2^, 6.14% ± 12.01%; Z = −2.047, *p* = 0.041) but decreased in the EXG (23.59 ± 4.70 to 21.53 ± 3.09 kg/m^2^, −7.34% ± 9.86%; Z = −2.432, *p* = 0.015) after 12 weeks, indicating a significant difference between groups (Z = −2.443, *p* = 0.014). On the other hand, muscle mass, as shown in Figure 4D, decreased in the COG (33.38 ± 5.41 to 29.94 ± 2.97 kg, −8.61% ± 13.58%; Z = −2.293, *p* = 0.022) but increased in the EXG (33.02 ± 7.10 to 35.64 ± 5.86 kg, 9.08% ± 8.79%; Z = −2.199, *p* = 0.028) after 12 weeks, indicating a significant difference between groups (Z = −2.698, *p* = 0.006). According to several studies, it is reported that exercise training brings desirable changes to body composition. In the results of this study, it can be seen that even ovarian cancer patients who had cancerous tissues showed similar results to those obtained in the general population if regular exercise was actively performed.

### 3.5. Positive Changes in Neutrophil and Lymphocyte Percentages by Exercise Training

As shown in Figure 5A, leucocytes showed no significant change in the COG (4.68 ± 0.71 × 10^2^ to 4.43 ± 0.59 × 10^2^ cells/µℓ, −3.89% ± 16.60%; Z = −1.511, *p* = 0.131) but increased in the EXG (4.74 ± 0.74 × 10^2^ to 5.69 ± 0.85 × 10^2^ cells/µℓ, 23.14% ± 27.58%; Z = −2.159, *p* = 0.031), indicating a significant difference between groups (Z = −3.550, *p* = 0.001). Neutrophil percentage for leucocytes, as shown in Figure 5B, was significantly decreased in the COG (47.33% ± 4.57% to 43.72% ± 2.58%, −6.95% ± 9.25%; Z = −2.329, *p* = 0.020) but increased in the EXG (49.77% ± 5.23% to 57.48% ± 7.28%, 16.06% ± 14.53%; Z = −2.746, *p* = 0.006), indicating a significant difference between groups (Z = −4.200, *p* = 0.001). Lymphocyte percentage for leucocytes, as shown in Figure 5C, was significantly decreased in the COG (38.61% ± 6.69% to 35.03% ± 4.07%, −7.60% ± 14.21%; Z = −2.017, *p* = 0.044) but increased in the EXG (39.76% ± 3.59% to 50.08% ± 5.91%, 27.23% ± 21.10%; Z = −2.906, *p* = 0.004), indicating a significant difference between groups (Z = −4.156, *p* = 0.001). These results suggest that regular and active exercise induced positive changes for neutrophils and lymphocytes, as well as white blood cells, which play a major role in human immunity.

### 3.6. Differences and Changes in NK Cells and NKT Cells for Lymphocytes by Exercise Training

As shown in Figure 6A, lymphocytes showed no significant change in the COG (15.13 ± 3.36 × 10^2^ to 13.13 ± 2.67 × 10^2^ cells/µℓ, −9.79% ± 22.46%; Z = −1.433, *p* = 0.152) but increased in the EXG (15.00 ± 2.26 × 10^2^ to 20.42 ± 2.11 × 10^2^ cells/µℓ, 38.23% ± 19.74%; Z = −2.968, *p* = 0.003), indicating a significant difference between groups (Z = −4.103, *p* = 0.001). NK cell percentage for lymphocytes, as shown in Figure 6B, was significantly decreased in the COG (7.99% ± 2.41% to 6.55% ± 2.05%, −8.01% ± 43.07%; Z = −0.938, *p* = 0.348) but increased in the EXG (8.65% ± 2.09% to 12.85% ± 3.44%, 51.63% ± 35.78%; Z = −2.835, *p* = 0.005), indicating a significant difference between groups (Z = −3.935, *p* = 0.001). NKT cell percentage for lymphocytes, as shown in Figure 6C, was not significantly changed in the COG (4.11% ± 2.64% to 3.03% ± 1.92%, −11.17% ± 40.88%; Z = −1.678, *p* = 0.093) and it was not significantly changed in the EXG (3.92% ± 1.82% to 5.46% ± 2.27%, 61.36% ± 84.04%; Z = −1.886, *p* = 0.059), indicating a significant difference between groups (Z = −2.860, *p* = 0.003).

### 3.7. The Changes of NK Cells’ Activator and Inhibitor

As shown in Figure 7A, total NK cells decreased in the COG (133.10 ± 28.65 to 103.10 ± 47.74 cells/µℓ, −21.23% ± 29.99%; Z = −2.273, *p* = 0.023), but increased in the EXG (133.58 ± 34.10 to 193.42 ± 70.80 cells/µℓ, 49.57% ± 56.97%; Z = −2.355, *p* = 0.019), indicating a significant difference between groups (Z = −3.345, *p* = 0.001). NKG_2_D+NK, as shown in Figure 7B, was not significantly changed in the COG (55.17% ± 12.06% to 46.36% ± 16.79%, −11.75% ± 36.74%; Z = −1.791, *p* = 0.073) but increased in the EXG (54.86% ± 11.40% to 66.72% ± 14.23%, 23.00% ± 22.94%; Z = −2.590, *p* = 0.010), indicating a significant difference between groups (Z = −2.978, *p* = 0.002). KIR_2_DL_3_+NK, as shown in Figure 7C, was significantly increased in the COG (34.35% ± 6.68% to 41.68% ± 5.35%, 24.33% ± 23.36%; Z = −3.354, *p* = 0.001) but decreased in the EXG (34.96% ± 12.55% to 24.18% ± 9.30%, −22.28% ± 34.46%; Z = −2.276, *p* = 0.023), indicating a significant difference between groups (Z = −3.883, *p* = 0.001).

### 3.8. Positive Changes in Cytotoxicity According to Exercise Training

Cytotoxicity, which indicates the cancer cell death rate of natural killer cells per unit of blood, is a method that can indirectly measure intrinsic immunity. As shown in Figure 8, the cytotoxicity was significantly decreased in the COG (2.70% ± 2.11% to 1.85% ± 1.17%, −17.42% ± 26.62%; Z = −2.766, *p* = 0.006) but increased in the EXG (2.98% ± 1.67% to 6.08% ± 2.49%, 137.38% ± 114.68%; Z = −2.668, *p* = 0.008), indicating a significant difference between groups (Z = −3.545, *p* = 0.001). These results suggest that the regular and active exercise program conducted in this study is a means to prevent potential cancer cell recurrence in ovarian cancer survivors.

## 4. Discussion

To our knowledge, this is the first study to provide evidence that a regular exercise program improves immunity by a positive change in body composition through enhanced physical fitness levels in ovarian cancer survivors. We found that a regular and active exercise program increases cardiorespiratory endurance, muscle strength and endurance, and flexibility, leading to healthy body composition and desirable changes in innate immunocytes variables.

When exercise starts, a parasympathetic nerve of the human body is suppressed, while a sympathetic nerve is activated, and movement becomes smooth. When the intensity or duration of exercise increases, the hypothalamic–pituitary–adrenal axis becomes more active, and various neurotransmitters, as well as hormones, are secreted more to the active muscle tissues [15]. Epinephrine secreted from the adrenal glands causes NK cells to be secreted from various tissues and is delivered throughout the body through blood and lymph circulation. A better physical condition, such as higher cardiorespiratory endurance or muscle mass, leads to better physiological or biochemical action. Obviously, the body’s immune system can exhibit different characteristics depending on aerobic exercise and resistance exercise. While some scholars claim that NK cells can be improved through aerobic exercise, other scholars argue that resistance exercise improves immune function by increasing the myokine according to muscle mass increase [38,39]. However, we tried to mix the optimal exercise factors that have been used for cancer survivors from the past until recently to increase their survival rate and improve their quality of life, considering that the subjects are patients with ovarian cancer removed. In this regard, Grote et al. investigated the efficacy of combined aerobic and resistance training for cancer survivors [40]. They reported that improving cardiometabolic health is critical for preventing comorbidity among cancer survivors—it is as important as monitoring for the possible recurrence of cancer [40]. Furthermore, Ligibel indicated the need to control for “energy balance factors, including diet, physical activity, and body weight” to reduce the risk of cancer recurrence following treatment [41]. In other words, as in previous studies, the results of this study imply that if the physical fitness level of cancer survivors is improved through exercise, the overall physical condition can be improved, thereby lowering the cancer recurrence rate.

Previous studies have suggested that positive or negative effects of physical exercise training appear differently in survivors of various types of cancer. Ovarian cancer has a low mortality rate and can be treated with various methods. It has consistently been reported that an exercise regimen should be part of a treatment program for ovarian cancer patients [1,42]. However, the optimal exercise regimens have not been specified for ovarian cancer patients. Therefore, this study was undertaken to investigate whether exercise training could increase physical fitness and improve immunity in ovarian cancer survivors. In relation to these results, all physical fitness factors investigated in this study showed significant differences after 12 weeks. When observing the results of this study shown in Figure 3, the cardiopulmonary endurance of the COG decreased by −8.89% ± 14.43% at week 12, while that of the EXG increased by 11.02% ± 9.28%. Although the delta% of flexibility, strength, and endurance in the COG decreased by −10.47% ± 23.60%, −10.12% ± 12.21%, and −22.13% ± 35.51%, respectively, those of the EXG significantly increased by 40.28% ± 27.86%, 28.32% ± 18.68%, and 126.54% ± 101.23%, respectively. These findings suggest that when ovarian cancer survivors perform exercise, positive changes occur in various physical factors, but among them, muscular endurance changes markedly. Changes in physical fitness level through exercise also cause desirable changes in body composition. When observing the results of this study shown in Figure 4, the body weight and fat mass of the COG increased by 4.58% ± 9.18% and 19.68% ± 26.97% at week 12, while those of the EXG decreased by −5.24% ± 5.98% and −8.78% ± 16.08%. Conversely, the muscle mass of the COG after 12 weeks decreased by −8.61% ± 13.58%, whereas that of the EXG significantly increased by 9.08% ± 8.79%. In the results of this study, it can be seen that even ovarian cancer patients who had cancerous tissues showed similar results to those obtained in the general population if regular exercise was actively performed. In other words, it was observed that “sarcopenia” appeared in the control group that did not perform the exercise, while “sarcomegaly” occurred in the exercise group that performed the exercise.

Regarding the above results, Pedersen reported that a moderate amount of exercise provides an overall “boost” to the immune system, but strenuous exercise results in dampening of the immune system. Strenuous exercise produces a remarkable increase in the levels of pro-inflammatory and anti-inflammatory cytokines, along with naturally occurring cytokine inhibitors and chemokines. That is, when the balance between pro-inflammatory and anti-inflammatory cytokines is disrupted, inflammation or pain can occur. In other words, seriously strenuous exercise provides a pro-inflammatory effect rather than an anti-inflammatory effect, which is more likely to cause diseases such as colds and flu [43,44,45]. Pedersen reported on the positive effects that moderate exercise has on both innate and acquired immunity, while Roessler et al. showed that specific immune cells are affected differently depending on the type of exercise [17,45]. According to Schneider et al. [46], cancer survivors respond differently to exercise depending on their location on the cancer continuum. However, when exercise is initiated, cancer survivors appear to effectively maintain or improve their physical and mental well-being. This can be a practical way to prevent the harmful toxic buildup from cancer treatments. Similar to this context, several previous studies do not seem to disagree that physical activity and/or exercise play an important role in innate immunocyte function [17,38,39,45,47].

Ovarian cancer refers to cancer in the ovaries, which play an essential role in female reproduction and hormone secretion. Cancer usually occurs in only one ovary and is bilateral in less than 10% of cases. In patients with ovarian cancer, physical exercise protects the body from cancer cells by improving physical fitness and immune function [9]. Physical exercise has a wide range of benefits during and after treatment that positively influence the quality of life in ovarian cancer survivors. These results could also be observed in our previous study [9]. In this pilot study, we found that the function of acquired immune cells was improved even when only resistance exercise that resulted in muscle hypertrophy was performed in ovarian cancer survivors. Del Giacco et al. indicated that an increase in NK cells enhances the cytolytic capacity for greater protection against diseases, while a decrease may indicate greater susceptibility to infections [48]. This is due to the sensitivity that NK cells have to the stress induced by physical exercise [49,50]. Similarly, the results of this study showed that NK cell levels tended to increase with regular exercise training. In other words, engaging in regular exercise increased the levels of NK cells in the peripheral blood and thus their availability to other tissues during physical exercise due to induced stress signals. Studies comparing the immunocytes of athletes and non-athletes generally indicate higher immunity in physically active individuals. In this context, the percentage of neutrophils to leucocytes in the COG of this study started to gradually decrease at week 12, whereas the neutrophil concentration of EXG increased at week 12. The percentage of lymphocytes to leucocytes also showed a similar tendency to that of neutrophils. It is associated to investigating the changes in immunocyte functions in participants who regularly train for a certain period with various intensities and durations under specific conditions. This study observed that the number of leukocytes, neutrophils, lymphocytes, and NK cells in the EXG was higher than in the COG. In addition, when the intensity of exercise was gradually increased, and the maximum level was reached, leukocytes, neutrophils, lymphocytes, and NK cells increased [9,48,49]. Similar to these results, the NK and NKT cell percentages of lymphocyte concentration showed positive changes in the EXG compared with the COG. When observing the results of this study, although the total NK and NKT cell percentages for lymphocytes decreased in the COG (−8.01% ± 43.07% vs. −11.17% ± 40.88%), they increased in the EXG (51.63% ± 35.78% vs. 61.36% ± 84.04%), indicating a boosted immune system. Moreover, in this study, we found that the positive improvement of NK and NKT cells in EXG was associated with a positive change in two receptors, namely, an increase in the NKG_2_D receptor and a decrease in the KIR_2_DL_3_ receptor. That is, this study demonstrated that the NKG_2_D receptor was upregulated for several NK cell-recruiting or NK cell-activating receptors, whereas the KIR_2_DL_3_ receptor was downregulated to maintain a high level of total NK cells. This means that a clear role for NK cells was shown in its antitumor effects due to long-term aerobic and resistance exercises. Similarly, studies by Gleeson [51] and Pedersen and Hoffman-Goetz [52] confirm the changing NK cell counts due to exercise. In fact, a lack of physical exercise during or after cancer treatment can exacerbate symptoms such as fatigue and contribute to loss of function, such as musculoskeletal and cardiovascular functions, hence contributing to the reduction in quality of life that comes with cancer treatment [53]. Eventually, this study revealed that regular exercise training can improve immune function by increasing the overall physical conditioning of cancer survivors. For future research studies, having a larger sample size with a more diverse demographic background would be encouraged, as well as observing a greater number of immunocytes. In addition, similar studies need to be described more thoroughly in the allocation/randomization of subjects in the future.

## 5. Conclusions

In this study, we found that long-term regular exercise improves the overall fitness of ovarian cancer survivors, leading to desirable changes in innate immunity. We also found that the cardiopulmonary and muscle endurance improve neutrophils, lymphocytes, NK cells, NKT cells, and receptors in ovarian cancer survivors. These results suggest that long-term regular exercise training for ovarian cancer survivors should be performed to increase the quality of life as well as the survival rate.

## Figures and Tables

**Figure 1 cancers-14-02808-f001:**
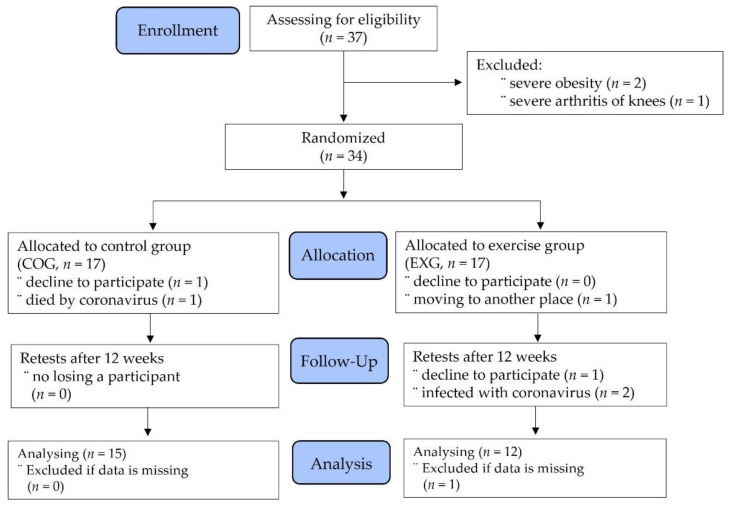
Participants’ allocation (consolidated standards for reporting of trials flow diagram).

**Figure 2 cancers-14-02808-f002:**
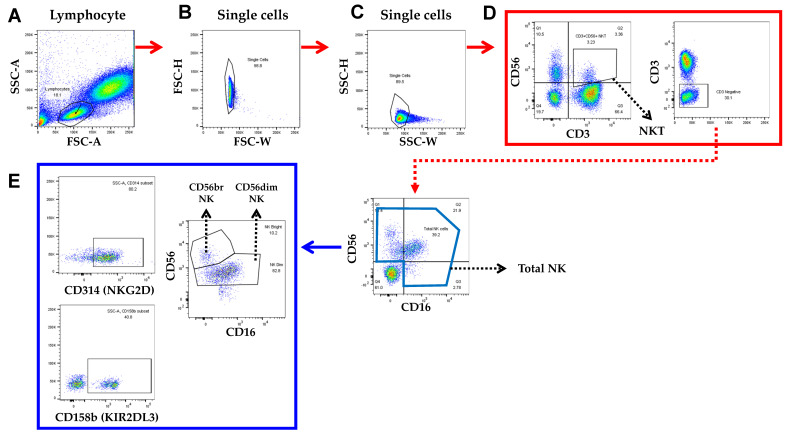
Flow cytometry analysis of NK cells in an ovarian cancer-removed survivor. A front scattered light (FSC)-A and side scattered light (SSC)-A plot were used to identify nucleated cells. (**A**) A lymphocyte gate was set based on the size and granularity of the cells. (**B**) Single cells were gated, and doublets were excluded. (**C**) CD14+ macrophages were excluded. (**D**) NK cells were defined as CD3−CD56+. (**E**) Two NK cell subsets were identified: CD16+ and CD16− NK cells. NKG2D and KIR2DL3 were defined as CD314 and CD158b, respectively.

**Figure 3 cancers-14-02808-f003:**
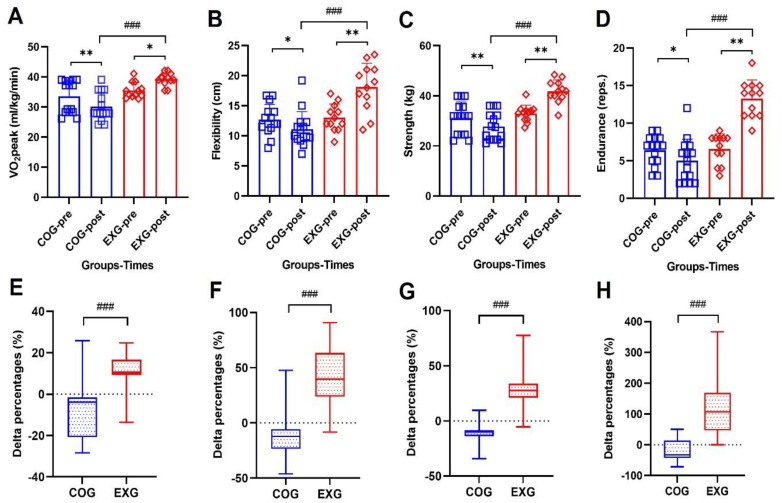
Aerobic exercise + resistance exercise for 6 days a week of 12 weeks improves the physical fitness of survivors after ovarian cancer resection. (**A**) Peak oxygen consumption (VO_2_ peak) in the control group (COG) and in the exercise group (EXG). (**B**) Flexibility in the COG and EXG. (**C**) Strength in the COG and EXG. (**D**) Muscle endurance in the COG and EXG. (**E**) Percentages of VO_2_ peak analyzed as Δ% in both groups. (**F**) Percentages of flexibility analyzed as Δ% in both groups. (**G**) Percentages of strength analyzed as Δ% in both groups. (**H**) Percentages of muscle endurance analyzed as Δ% in both groups. All data are represented as means ± SD. *p*-values were calculated by Student’s *t* test or Mann–Whitney U test. For (**A**) to (**H**), these tests were performed independently at each time point; ^###^
*p* < 0.001. *p*-values were calculated by Student’s *t* test or Wilcoxon signed rank test. For (**A**) to (**D**), these tests were performed dependently at each time point; * *p* < 0.05 and ** *p* < 0.01.

**Figure 4 cancers-14-02808-f004:**
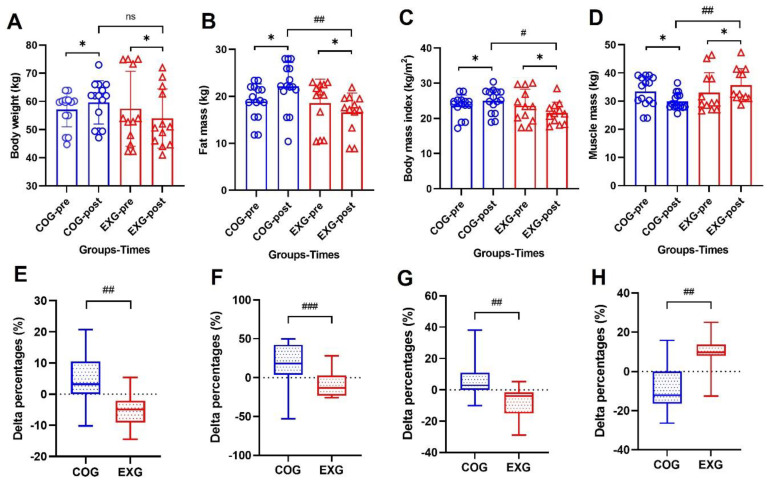
Body composition is improved in the ovarian cancer survivors after 12 weeks of aerobic and resistance exercise. (**A**) Body weight in the control group (COG) and in the exercise group (EXG). (**B**) Fat mass in the COG and EXG. (**C**) Body mass index in the COG and EXG. (**D**) Muscle mass in the COG and EXG. (**E**) Percentages of body weight analyzed as Δ% in both groups. (**F**) Percentages of fat mass analyzed as Δ% in both groups. (**G**) Percentages of body mass index analyzed as Δ% in both groups. (**H**) Percentages of muscle mass analyzed as Δ% in both groups. All data are represented as means ± SD. *p*-values were calculated by Student’s *t* test or Mann–Whitney U test. For (**A**) to (**H**), these tests were performed independently at each time point; ns: not significance; ^#^
*p* < 0.05, ^##^
*p* < 0.01, and ^###^
*p* < 0.001. *p*-values were calculated by Student’s *t* test or Wilcoxon signed rank test. For (**A**) to (**D**), these tests were performed dependently at each time point; * *p* < 0.05.

**Figure 5 cancers-14-02808-f005:**
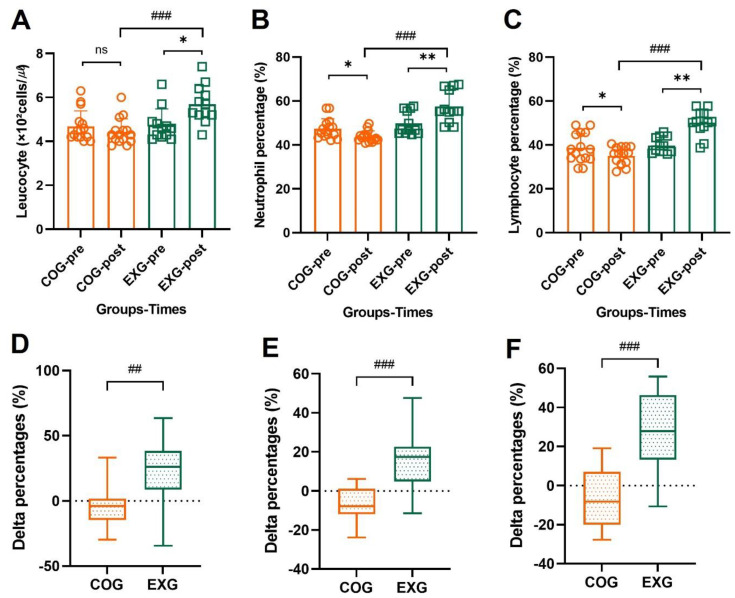
Neutrophils and lymphocytes of leucocytes are improved in the ovarian cancer survivors after 12 weeks of aerobic and resistance exercise. (**A**) Leucocyte’s absolute values in the control group (COG) and in the exercise group (EXG). (**B**) Neutrophil percentages of leucocyte’s absolute values in the COG and EXG. (**C**) Lymphocyte percentages of leucocyte’s absolute values in the COG and EXG. (**D**) Percentages of leucocyte analyzed as Δ% in both groups. (**E**) Percentages of neutrophil analyzed as Δ% in both groups. (**F**) Percentages of lymphocyte analyzed as Δ% in both groups. All data are represented as means ± SD. *p*-values were calculated by Student’s *t* test or Mann–Whitney U test. For (**A**) to (**F**), these tests were performed independently at each time point; ns: not significance; ^##^
*p* < 0.01, and ^###^
*p* < 0.001. *p*-values were calculated by Student’s *t* test or Wilcoxon signed rank test. For (**A**) to (**C**), these tests were performed dependently at each time point; * *p* < 0.05 and ** *p* < 0.01.

**Figure 6 cancers-14-02808-f006:**
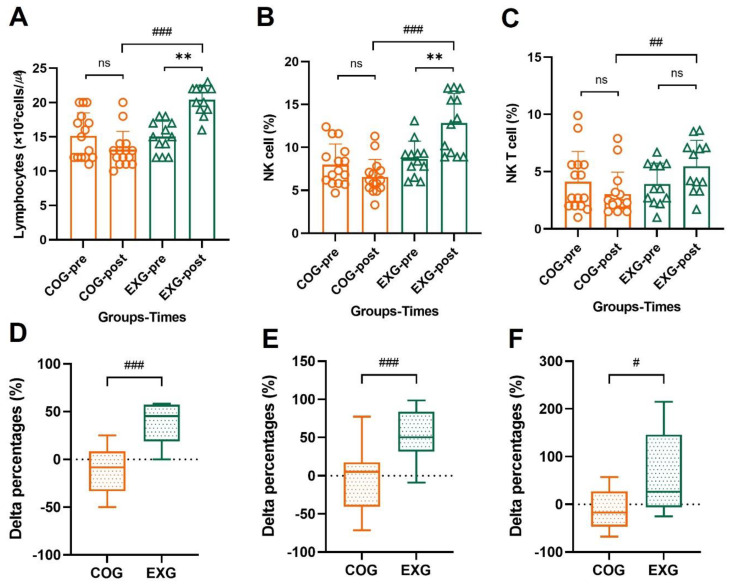
NK and NKT cells of lymphocytes are improved in the ovarian cancer survivors after 12 weeks of aerobic and resistance exercise. (**A**) Lymphocyte’s absolute values in the control group (COG) and in the exercise group (EXG). (**B**) NK cell percentages of lymphocyte’s absolute values in the COG and EXG. (**C**) NKT cell percentages of lymphocyte’s absolute values in the COG and EXG. (**D**) Percentages of lymphocyte analyzed as Δ% in both groups. (**E**) Percentages of NK cells analyzed as Δ% in both groups. (**F**) Percentages of NKT cells analyzed as Δ% in both groups. All data are represented as means ± SD. *p*-values were calculated by Student’s *t* test or Mann–Whitney U test. For (**A**) to (**F**), these tests were performed independently at each time point; ^#^
*p* < 0.05, ^##^
*p* < 0.01, and ^###^
*p* < 0.001. *p*-values were calculated by Student’s *t* test or Wilcoxon signed rank test. For (**A**) to (**C**), these tests were performed dependently at each time point; ns: *p* > 0.05 and ** *p* < 0.01.

**Figure 7 cancers-14-02808-f007:**
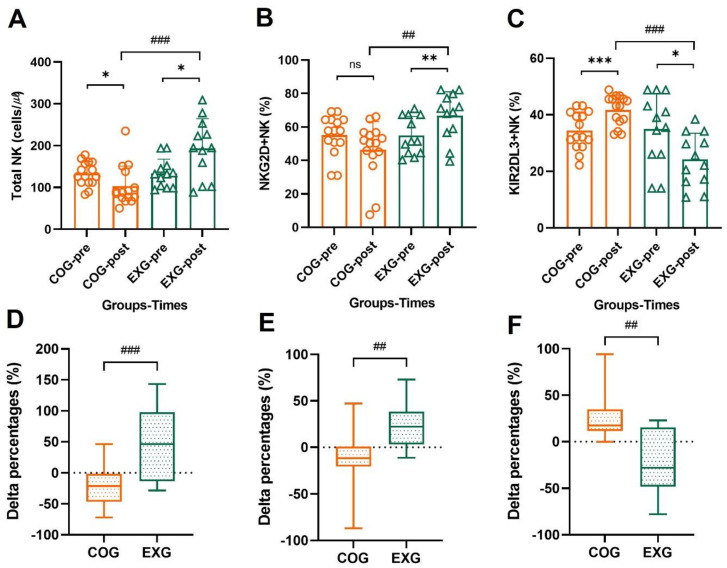
NKG_2_D+NK and KIR_2_DL_3_+NK receptors of total NK cells are improved in the ovarian cancer survivors after 12 weeks of aerobic and resistance exercise. (**A**) Total NK cells’ absolute values in the control group (COG) and in the exercise group (EXG). (**B**) NKG_2_D+NK receptor percentages of total NK cells’ absolute values in the COG and EXG. (**C**) KIR_2_DL_3_+NK receptor percentages of total NK cells’ absolute values in the COG and EXG. (**D**) Percentages of total NK cells analyzed as Δ% in both groups. (**E**) Percentages of NKG_2_D+NK receptor analyzed as Δ% in both groups. (**F**) Percentages of KIR_2_DL_3_+NK receptor analyzed as Δ% in both groups. All data are represented as means ± SD. *p*-values were calculated by Student’s *t* test or Mann–Whitney U test. For (**A**) to (**F**), these tests were performed independently at each time point; ^##^
*p* < 0.01 and ^###^
*p* < 0.001. *p*-values were calculated by Student’s *t* test or Wilcoxon signed rank test. For (**A**) to (**C**), these tests were performed dependently at each time point; ns: *p* > 0.05, * *p* < 0.05, ** *p* < 0.01, and *** *p* < 0.001.

**Figure 8 cancers-14-02808-f008:**
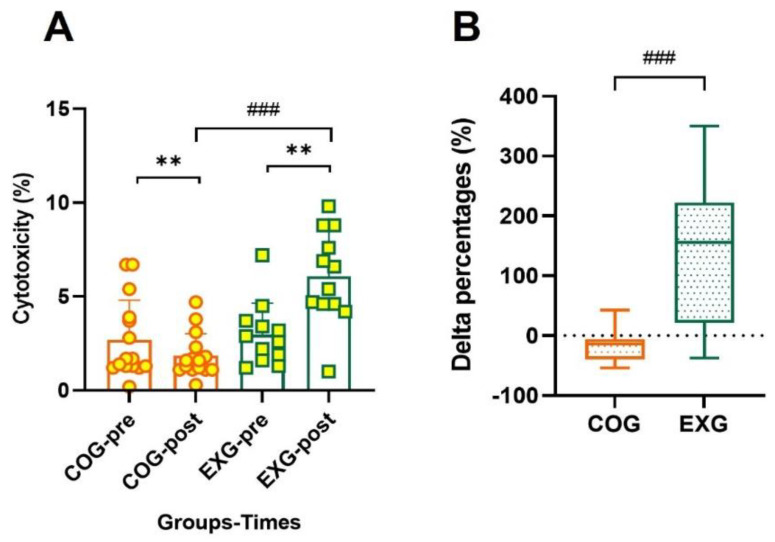
Cytotoxicity increases in the ovarian cancer survivors after 12 weeks of aerobic and resistance exercise. (**A**) Cytotoxic values in the control group (COG) and in the exercise group (EXG). (**B**) Percentages of cytotoxicity analyzed as Δ% in both groups. All data are represented as means ± SD. *p*-values were calculated by Student’s *t* test or Mann–Whitney U test. For (**A**) to (**B**), these tests were performed independently at each time point; ^###^
*p* < 0.001. *p*-values were calculated by Wilcoxon signed rank test. For (**A**), the test was performed dependently at each time point; ** *p* < 0.01.

**Table 1 cancers-14-02808-t001:** The exercise training programs for ovarian cancer survivors.

Items	Exercise Types		Exercise Training Phases			
			Phase I	Phase II	Phase III	Phase IV
			0–3 Week	4–6 Week	7–9 Week	10–12 Week
Warm-up	Standing stretching		10 min	10 min	10 min	10 min
Work-out	Aerobic training	F	3 days	3 days	3 days	3 days
		I	40% VO_2_ peak	50% VO_2_ peak	60% VO_2_ peak	70% VO_2_ peak
		T	50 min	45 min	40 min	35 min
	Resistance training	F	3 days	3 days	3 days	3 days
		I	12RM × 3 sets	10RM × 3 sets	8RM × 3 sets	6RM × 3 sets
		T	30–40 min	30–40 min	30–40 min	30–40 min
Cool-down	Sitting/Lying stretching		15 min	15 min	15 min	15 min

F, frequency; I, intensity; T, time; RM, repetition maximum.

**Table 2 cancers-14-02808-t002:** Demographic and clinical characteristics of the ovarian cancer survivors.

Items	Groups			
	Total (*n* = 27)	COG (*n* = 15)	EXG (*n* = 12)	*p*
Age (y)	51.07 ± 5.68	50.60 ± 5.83	51.67 ± 5.68	0.548
Height (cm)	155.74 ± 4.08	155.80 ± 3.10	155.67 ± 5.21	0.981
Body weight (kg)	57.26 ± 9.76	57.14 ± 6.13	57.41 ± 13.33	0.486
Body mass index (kg/m^2^)	23.60 ± 3.78	23.61 ± 3.03	23.59 ± 4.70	0.829
Married, *n* (%)	23 (85.2%)	13 (86.7%)	10 (83.3%)	0.905
Employed, *n* (%)	8 (29.6%)	6 (40.0%)	2 (16.7%)	0.323
Current smoker, *n*	0 (0.0%)	0 (0.0%)	0 (0.0%)	-
Medications’ number	1.85 ± 0.99	1.87 ± 1.13	1.83 ± 0.83	0.867
Supplements’ number	1.89 ± 0.64	1.87 ± 0.64	1.92 ± 0.67	0.867
Comorbidities’ number	2.58 ± 0.97	2.40 ± 0.98	2.83 ± 0.94	0.236
Ovarian cancer diagnosed year	4.06 ± 0.49	4.12 ± 0.47	3.99 ± 0.52	0.373
Primary cytoreductive surgery (month)	46.40 ± 5.93	46.18 ± 6.00	46.67 ± 6.20	1.000
Primary cytoreductive surgery, *n* (%)	20 (74.1%)	11 (73.3%)	9 (75.0%)	-
Interval debulking surgery (month)	44.71 ± 6.02	43.75 ± 5.62	46.00 ± 7.55	0.857
Interval debulking surgery, *n* (%)	7 (25.9%)	4 (26.7%)	3 (25.0%)	-
Lymph node metastasis, *n* (%)	5 (18.5%)	2 (13.3%)	3 (25.0%)	0.614
Organ metastasis, *n* (%)	3 (11.1%)	1 (6.7%)	2 (16.7%)	0.683
Other ovarian cancer-related treatment				-
Surgery, *n* (%)	22 (81.5%)	12 (80.0%)	10 (83.3%)	0.905
Radiation therapy, *n* (%)	17 (63.0%)	10 (66.7%)	7 (58.3%)	0.719
Chemotherapy, *n* (%)	22 (81.5%)	12 (80.0%)	10 (83.3%)	0.905

All data represent mean ± standard deviation or number (%). COG, control group; EXG, exercise group.

**Table 3 cancers-14-02808-t003:** Changes and differences in mean daily calorie intake and mean daily calorie output.

Items	Time	Groups		
		COG (*n* = 15)	EXG (*n* = 12)	*p*
Daily calorie (kcal/day)	pre	1618.93 ± 187.75	1591.25 ± 166.90	0.829
	post	1732.00 ± 263.51	1735.50 ± 152.86	0.581
Carbohydrate (g/day)	pre	117.07 ± 15.50	119.83 ± 12.42	0.581
	post	118.33 ± 13.91	117.50 ± 13.01	0.792
Fiber (g/day)	pre	19.40 ± 2.82	19.17 ± 4.91	0.755
	post	20.80 ± 2.98	20.08 ± 4.46	0.867
Protein (g/day)	pre	50.20 ± 6.28	51.17 ± 6.95	0.905
	post	51.67 ± 5.61	50.67 ± 7.88	0.581
Fat (g/day)	pre	51.93 ± 7.56	50.17 ± 9.03	0.648
	post	50.80 ± 7.28	50.67 ± 9.75	0.981
Vitamin A (ug RAE/day)	pre	538.13 ± 66.31	550.42 ± 62.13	0.792
	post	549.13 ± 59.58	549.92 ± 55.70	0.792
Vitamin C (mg/day)	pre	94.40 ± 13.39	94.83 ± 12.96	0.829
	post	98.13 ± 14.27	95.67 ± 17.07	0.648
Calcium (mg/day)	pre	706.20 ± 112.59	690.58 ± 84.11	0.683
	post	687.73 ± 137.01	680.08 ± 137.37	0.981
Iron (mg/day)	pre	12.87 ± 3.46	11.42 ± 2.71	0.300
	post	12.40 ± 3.54	11.58 ± 2.39	0.683
Daily physical activity	pre	264.47 ± 27.83	276.00 ± 10.37	0.516
(kcal/day)	post	286.00 ± 31.89	283.92 ± 30.30	0.792

All data represent mean ± standard deviation. COG, control group; EXG, exercise group.

## Data Availability

The data presented in this study are available on request from the corresponding author.

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
