# Peer review of "Immunoprotecting Effects of Exercise Program against Ovarian Cancer: A Single-Blind, Randomized Controlled Trial"

_cancers, 2022, doi:10.3390/cancers14112808_

Round 1
Reviewer 1 Report
General comments:
The present manuscript deals with an important topic, which concerns patients suffering from ovarian cancer. The authors performed an interesting pilot study among ovarian cancer survivors investigating the effect of an exercise intervention on several aspects of immunoprotecting effects of aerobic and resistance exercise, as well. The present manuscript is well written. It allows important insights in this topic. The methods were appropriate, and the results as well as the discussion are written in a clear manner. Nevertheless, there seem to be few limitations, which should addressed.
Special comments:
- Title:
A randomized controlled trials: please change to a randomized controlled trial
- Please clarify which information you can add to the existing scientific literature.
- What can be done in daily routine to improve knowledge about exercise? Are your effects due to strength exercise or aerobic exercise?
- Please update your reference list.
- Please describe the exercise intervention more in detail.
Author Response
Answers to reviewer’s comments
Thank you for your kind advice and comments for publication in Cancers. We revised our manuscript as per your comments. We represented the specific modifications in response to the comments by blue letters in our manuscript. We sincerely appreciate your comments because your comments make our manuscript better.
Reviewer 1
General comments:
The present manuscript deals with an important topic, which concerns patients suffering from ovarian cancer. The authors performed an interesting pilot study among ovarian cancer survivors investigating the effect of an exercise intervention on several aspects of immunoprotecting effects of aerobic and resistance exercise, as well. The present manuscript is well written. It allows important insights in this topic. The methods were appropriate, and the results as well as the discussion are written in a clear manner. Nevertheless, there seem to be few limitations, which should addressed.
Special comments:
#Q1. Title: A randomized controlled trials: please change to a randomized controlled trial
#Response 1: Thank you for what the reviewer has pointed out the comment. We have corrected the point you pointed out as follows.
A randomized controlled trials → A single-blind, randomized controlled trial
#Q2. Please clarify which information you can add to the existing scientific literature.
#Response 2: Thank you for what the reviewer has pointed out the comment. We have inserted the reference on Line 44, Line 53, and Line 78 as follows.
Line 44 → [3] Guo, J.; Yang, W.L.; Pak, D.; Celestino, J.; Lu, K.H.; Ning, J.; Lokshin, A.E.; Cheng, Z.; Lu, Z.; Bast, R.C. Jr. Osteopontin, Macrophage Migration Inhibitory Factor and Anti-Interleukin-8 Autoantibodies Complement CA125 for Detection of Early Stage Ovarian Cancer. Cancers (Basel). 2019, 11(5), 596. doi: 10.3390/cancers11050596.
Line 53 → [7] Pin, F.; Barreto, R.; Kitase, Y.; Mitra, S.; Erne, C.E.; Novinger, L.J.; Zimmers, T.A.; Couch, M.E.; Bonewald, L.F.; Bonetto. A. Growth of ovarian cancer xenografts causes loss of muscle and bone mass: a new model for the study of cancer cachexia. J Cachexia Sarcopenia Muscle. 2018, 9, 685-700. doi: 10.1002/jcsm.12311.
Line 78 → [15] Smith, S.M.; Vale, W.W. The role of the hypothalamic-pituitary-adrenal axis in neuroendocrine responses to stress. Dialogues Clin Neurosci. 2006, 8, 383-395. doi: 10.31887/DCNS.2006.8.4/ssmith.
#Q3. What can be done in daily routine to improve knowledge about exercise? Are your effects due to strength exercise or aerobic exercise?
#Response 3: Thank you for what the reviewer has pointed out the comment. The answer to your question is below. And, blue sentences are added on line 522 and below.
“Obviously, the body's immune system can exhibit different characteristics depending on aerobic exercise and resistance exercise. While some scholars claim that NK cells can be improved through aerobic exercise, some scholars argue that resistance exercise improves immune function by increasing the myokine according to muscle mass increase. However, we tried to mix the optimal exercise factors that have been used for cancer survivors from the past until recently in order to increase their survival rate and improve their quality of life, considering that the subjects are patients with ovarian cancer removed.” Therefore, in this study, it seems that there is a limit to which type of exercise is more beneficial to ovarian cancer survivors.
Q4. Please update your reference list.
#Response 4: Thank you for what the reviewer has pointed out the comment. We have updated our references and inserted recent references as follows.
Line 526 → [38] Jee, Y.S. Cancer and exercise immunity: 6th series of scientific evidence. J. Exerc. Rehabil. 2021, 17, 151-152. doi: 10.12965/jer.2142274.137.
Line 526 → [39] Jee, Y.S. Exercise-induced myokines for cancer patients: 7th in a series of scientific evidence J. Exerc. Rehabil. 2021, 17, 293-294. doi: 10.12965/jer.2142560.280.
Line 581 → [47] Jee, Y.S. Exercise rehabilitation strategy for the prevention of sarcopenia in cancer populations: 8th in a series of scientific evidence. J. Exerc. Rehabil. 2022, 18, 79-80. doi: 10.12965/jer.2244124.062.
Q5. Please describe the exercise intervention more in detail.
#Response 5: Thank you for what the reviewer has pointed out the comment. We added additional explanations for exercise interventions as follows and inserted those sentences on Line 278.
When designing an exercise program, we prescribed 150 min of moderate-intensity or 75 min of vigorous-intensity exercise per week, as suggested by the American Cancer Society and the American College of Sports Medicine as shown in Line 254 to 257. Especially, the participants of EXG began with warm-up conditioning for 10 min, as instructed by the researcher, and then performed the workout. In the case of aerobic exercise training, participants performed aerobic exercise 3 days a week (Mondays, Wednesdays, and Fridays), and the intensity of exercise was walking or running at 40–70% of the peak oxygen uptake. A heart monitor was used to maintain heart rate according to the exercise intensity. The exercise started with 50 min of walking in Phase I and 35 min of brisk walking to light jogging in Phase IV, after which the program ended. This study was also applied to ovarian cancer survivors with an acceptable resistance exercise from 12 repetitions maximum (RM) to 6 RM that could cause muscle hypertrophy based on previous studies [7,30,31]. EXG participated in a resistance exercise for 3 days per week (Tuesdays, Thursdays, and Saturdays). For the safety of the ovarian cancer survivors, exercise using weight machines was prescribed, and according to exercise order, multiple-joint exercises centered on large muscles were performed. Afterward, single-joint exercises were carried out in parallel. The order of resistance exercise consisted of lateral pull-downs, chest presses, leg presses, abdominal crunches, and back extensions on Tuesdays; seated cable rows, chest butterflies, deadlifts, leg extensions, and leg curls on Thursdays; and lateral pulldowns, chest presses, leg presses, abdominal crunches, and back extensions on Saturdays. The reason why the exercise was prescribed differently for every other day was to ensure that the patients had no experience in exercising regularly and that the exercise they encountered for the first time did not put any strain on the muscles and tendons. All exercises were performed under strict supervision, and a rating of perceived exertion was applied to each exercise to identify the fatigue from the exercise and then adjust the rest time.
Sincerely,
Thank you for your comments, we represented the modifications in response to your comments.
May 24, 2024

Reviewer 2 Report
Dear authors
I've no issues about the article; methods and results are well described, RCT was conducted in the respect of Consort guidelines. The only suggestion for future study is to better manage allocation/randomization: a software is even more reliable to this scope.
Author Response
Answers to reviewer’s comments
Thank you for your kind advice and comments for publication in Cancers. We revised our manuscript as per your comments. We represented the specific modifications in response to the comments by blue letters in our manuscript. We sincerely appreciate your comments because your comments make our manuscript better.
Reviewer 2
I've no issues about the article; methods and results are well described, RCT was conducted in the respect of Consort guidelines. The only suggestion for future study is to better manage allocation/randomization: a software is even more reliable to this scope.
#Response 1: Thank you for what the reviewer has pointed out the comment. In our future research, we will do our best to make a high-quality thesis by thorough allocation/ randomization of the subjects. According to your opinion, 'In addition, similar studies need to be described more thoroughly in the allocation/randomization of subjects in the future.' is inserted on line 627.
Sincerely,
Thank you for your comments, we represented the modifications in response to your comments.
May 24, 2024

Reviewer 3 Report
The study is found to be good in the context of effect of exercises among ovarian cancer survivors. However few points noted are mentioned below;
- Title could be reframed especially in terms of the study design
- Subjects allocation could be explained in a better way
- Even though, the experimental group had given the exercise programs, the control group interventions needs clarity.
Author Response
Answers to reviewer’s comments
Thank you for your kind advice and comments for publication in Cancers. We revised our manuscript as per your comments. We represented the specific modifications in response to the comments by blue letters in our manuscript. We sincerely appreciate your comments because your comments make our manuscript better.
Reviewer 3
The study is found to be good in the context of effect of exercises among ovarian cancer survivors. However few points noted are mentioned below;
Q1. Title could be reframed especially in terms of the study design
#Response 1: Thank you for what the reviewer has pointed out the comment. Based on your point, the title has been modified as follows.
“Immunoprotecting effects of exercise program against ovarian cancer: A single-blind, randomized controlled trial”
Q2. Subjects allocation could be explained in a better way
#Response 2: Thank you for what the reviewer has pointed out the comment. Based on your point, the Subjects allocation method has been added on Line 117 as follows.
“They continued to receive standard care, with the agreement that existing physical activity was to remain unchanged over the duration of the trial. Participants were informed of their allocation by a researcher and were randomly allocated to either the control group (COG, n = 15) or exercise group (EXG, n = 12). At any stage, all assessments were done at sites from the hospital, and that participants were reminded before each assessment not to disclose their allocation.”
Q3. Even though, the experimental group had given the exercise programs, the control group interventions needs clarity.
#Response 3: Thank you for what the reviewer has pointed out the comment. Based on your point, the control group interventions were modified as follows and inserted those sentences on Line 138.
“As opposed to EXG participating in exercise, COG patients gathered at the same center at a scheduled time and were allowed to meditate in a lying position on a mattress for 60 min. At this time, the COG was allowed to lightly stretch in a lying position on the floor, but did not engage in strength training or other aerobic exercise, while sleeping when they were drowsy.
Sincerely,
Thank you for your comments, we represented the modifications in response to your comments.
May 24, 2024
